# WHEN AND WHERE DO FEED-FORWARD NEURAL NETWORKS LEARN LOCALIST REPRESENTATIONS?

## ABSTRACT

According to parallel distributed processing (PDP) theory in psychology, neural networks (NN) learn distributed rather than interpretable localist representations. This view has been held so strongly that few researchers have analysed single units to determine if this assumption is correct. However, recent results from psychology, neuroscience and computer science have shown the occasional existence of local codes emerging in artificial and biological neural networks. In this paper, we undertake the first systematic survey of when local codes emerge in a feed-forward neural network, using generated input and output data with known qualities. We find that the number of local codes that emerge from a NN follows a well-defined distribution across the number of hidden layer neurons, with a peak determined by the size of input data, number of examples presented and the sparsity of input data. Using a 1-hot output code drastically decreases the number of local codes on the hidden layer. The number of emergent local codes increases with the percentage of dropout applied to the hidden layer, suggesting that the localist encoding may offer a resilience to noisy networks. This data suggests that localist coding can emerge from feed-forward PDP networks and suggests some of the conditions that may lead to interpretable localist representations in the cortex. The findings highlight how local codes should not be dismissed out of hand.

## 1 INTRODUCTION AND RELATED WORK

Local neural network models, which are often argued to be biologically implausible, have nevertheless been built or discussed by psychologists such as McClelland & Rumelhart (1981); Rumelhart & McClelland (1982); Page (2000), and a few researchers like Berkeley et al. (1995) have done single neuron probing studies (the equivalent of the neuroscience approach) on their neural networks. However, as parallel distributed processing (PDP) neural networks (NN), as discussed by Rumelhart et al. (1986a;b) and Rogers & McClelland (2014), are generally assumed to learn distributed encodings across all situations, it is often believed that a single neuron in an artificial neural network is not interpretable, and experiments to test if this is true are rarely performed.

Recently, however, there has been evidence emerging from neuroscience and modern artificial neural networks that demonstrate the existence of interpretable, local codes. Marr et al. (1991) argued that the neurons in the hippocampus codes for information in a highly selective manner in order to learn quickly without forgetting (catastrophic interference), and Bowers et al. (2014) argued that some neurons in cortex are highly selective in order to encode multiple items at the same time in short-term memory (solving the so-called superposition catastrophe). Quiroga et al. (2005) reported single cells that fire frequently in response to one stimulus, which suggests that individual neurons can be usefully interpreted.

Localist codes have been found in artificial neural networks, see Bowers (2017; 2009) for full reviews, some examples are Le (2013); Gubian et al. (2017)). Bowers et al. (2014) have shown that PDP models learn localist codes when trained to co-activate multiple items at the same time. Deep networks learn selective codes under some conditions. For example, Karpathy et al. (2015)'s found quote mark detectors in RNNs. And there is also some evidence from feed-forward models. For example, Nguyen et al. (2016) have found that probing individual hidden layer neurons (HLNs) with noise and using activation maximisation they can produce a picture of what that neuron will respond

most to, and from this they identified HLNs that act as feature detectors, such as those that only responding to creases (in clothing) or eyes or faces and so on.

We would like to elucidate the conditions in which simple networks learn selective units, as this may provide further insight into the conditions in which neurons in cortex respond selectively, and, as we expect such codes are learned for sound information theoretic reasons, we expect that these conditions will also apply to when neural networks might learn them as well. Thus, in this paper, we undertake a study of simple feed-forward neural networks to investigate whether local codes, LCs, do actually emerge in PDP networks, and (as we shall show that they do) we then look at what inhibits or promotes the emergence of LCs by designing input and output data with known properties. And, as this data is structured to have some invariance within a class and some randomness, it is proposed that these experiments could as be modelling the layers within the deep neural network above those which transform the input data from pixel space to feature space.

To be clear, we consider a neuron to be interpretable if probing of the activation state of it could give correct and useful information about the classification of the input. We look for information about the presence or absence of a category in the hidden layer (category selective HLNs). We separate the qualitative measure (selective) of whether or not a HLN encodes category presence, from the quantitative measure of how much the HLN responds to a category (selectivity). Thus, a HLN is selective if it encodes the presence/absence of a category. Examples are shown in figure 1, as the neuron encodes the presence or absence of the category shown as red circles, equivalently, it could be claimed that these neurons are selective for that category. As biological neurons use energy to encode information, a selective neuron is usually 'selectively on' (see figure 1(left)), but as there is no energy cost in neural networks 'selectively off' units (figure 1(right)) have also been observed. We use the word 'selectivity' as a quantitative measure of the difference between activations for the two categories, A and not-A, where A is the category a neuron is selective for (and not-A being all other categories). Specifically:

$$\text{selectivity} = \text{Min}[\text{Min}[A] - \text{Max}[\neg A], \ \text{Max}[A] - \text{Min}[\neg A]] \ . \tag{1}$$

The important point is the qualitative measure of whether or not a neuron is selective, not how much it is selective by, as we are interested in counting the number of local codes that emerge. Note that the chance that all the members of A would emerge disjoint from the members of not-A is $\binom{50}{50}/\binom{500}{50}$ is tiny ($4.32 \times 10^{-71}$). Furthermore, we found that the selectivity increased with training as the neural network minimised the loss function, but that the number of selective codes did not change once the neural network achieved 100% accuracy.

A criticism often made of a grandmother cell hypothesis is that even if a cell fires consistently to a single class, it is not possible to know that it would not have fired to a stimulus that was not presented. For example, although Quiroga et al. (2005) found a neuron that responded selectively to images of Jennifer Aniston, the authors only presented approximately 100 images to the human participant, and it is possible that other non-tested images would also drive the neuron. Waydo et al. (2006) estimated that between 50-150 other images would drive this neuron. Obviously an experimenter cannot present every possible combination of visual inputs to a patient. However, in neural networks with small datasets, we can present all the possible stimuli to the network. We consider a neuron to be selective if it is selective over all the data it is reasonable to expect the network to differentiate between. For example, it is reasonable to do the test over all training data, and it is reasonable to do it over all test and all verification data or even other data of a similar form (such as different photos of the same class), and choosing what constitutes a reasonable set of data is a decision to be made by the experimenters and reviewers. In this work, we chose to use a simple pattern classification task, rather than an image recognition task, as we could then test the NN with all possible patterns.

## 2 METHODOLOGY

**Data design**   Data input to a neural network can be understood as a code, $\{C_x\}$, with each trained input data vector designated as a codeword, $C_x$. The size of the code is related to the number of codewords (i.e. the size of the training set), $n_x$. $L_x$ is the length of the codeword, and is 500bits in this paper. We used a binary alphabet, and the number of '1's in a codeword is the weight, $w_x$ of that codeword (this weight definition is not the same as connection weights in the neural network).

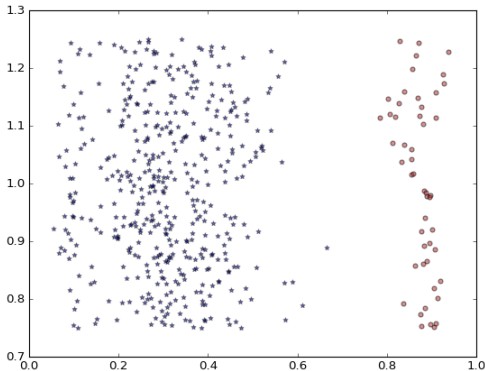 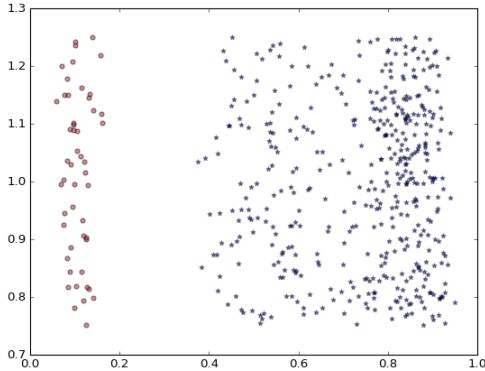

Figure 1: Examples of interpretable local codes found in a distributed network. Left: a selectively on unit with a selectivity of $\sim +0.12$; Right: selectively off unit with a selectivity of $\sim -0.2$. Red circles belong to a single category, blue stars are all the members of all other categories, the x-axis is the activation of a hidden layer neuron (HLN) and points are jittered randomly around 1 on the y-axis for ease of viewing. There is a clear separation between activations for the class depicted in red (A) and all other activations (not-A), thus examination of the activations of these units would reveal the presence or absence of the red class.

To create a set of $n_P$ classes with a known structural similarity, the procedure in figure 2 was followed. We start with a set of $n_P$ prototypes, $\{P_x, 1 \leq x \leq n_P\}$, with blocks of '1's of length $L_P/n_P$, called prototype blocks, which code for a class. For example, if $L_x$ were 12 and $n_P$ were 3: $P_1 = [111100000000]$ and $P_2 = [000011110000]$, $P_3 = [000000001111]$, and this would gives prototypes that are a Hamming distance of 8 apart (Hamming distance is the number of bits which must be switched to change one codeword into another), and thus we know that our prototypes span input-data space. To create members of each class, the prototype is used as a mask, with the '0' blocks replaced by blocks from a random vector, $R_x$. The weight of the random vectors, $w_R$ can be tuned to ensure that a set of vectors randomly clustered around the prototype vector are generated, such that members of the same category are closer to each other than those of the other categories (N.B. the prototypes are not included as members of the category). A more realistic dataset is created by allowing the prototypes to be perturbed so that a percentage of the prototype block is randomly switched to '0's each time a new codeword is created, in accordance with the perturbation rate (see $P_2'$ in figure 2). This method creates a code with a known number of invariant bits (bits that are the same) between codewords in the same category. For example, in figure 2, codewords $C_1$ and $C_2$ were both derived from $P_1$, and have a Hamming distance of 6, where as $C_1$ and $C_4$ are in different classes and have a Hamming distance of 8. Note that the difference between these numbers is much bigger in our experiments as we used vectors of length 500 split into 10 categories). We define 'sparseness' of a vector, $S_x$, as the fraction of bits that are '1's. Furthermore, local codes were highly unlikely to appear in the input codes and we did not observe any in a few random checks.

**Neural network design** The neural networks are three-layer feed-forward networks, with $L_x = 500$ input neurons, $n_{HLN}$ hidden layer neurons (HLN) and either 50 or 10 output neurons. All input vectors were 500bits long and were matched to 10 output classes, with the output encoded either as a 50bit long distributed vector (with weight 25) or a 10bit long 1-hot output vector. These networks are intended to also model single layers within a deeper and more complex network, and thus no form of softmax activation on the output was used. For most of the data, we chose to use sigmoidal activation functions for ease of understanding (activation is strictly limited between 0 and 1), but ReLU neurons were also tested. All networks had 100% accuracy on the task (with an output of more than 0.9 being taken as a one and less than 0.1 as a zero, although the networks were closer than these limits at the end of training). The networks were set-up in Keras (Chollet (2015)) using a Tensorflow (Abadi et al. (2015)) back-end, and run on an Titan X Nvidia GPU under Ubuntu Linux. Each plotted data point comes from an average of at least 10 trained networks, with

error bars calculated as the standard error from the measured data. Neural networks were trained for 45,000 epochs. Neurons were counted as a local code if the selectivity was above 0.05, most neurons were higher. Unless stated differently in the experiment, neural networks were run with sigmoidal neurons, 500 500bit long vectors separated into 10 classes, with 50bit long distributed output codes, no perturbation of the prototype block code and no dropout applied (these are the conditions for the black-dashed data in figures 3, 4 and 6 and this dataset is used as our standard to compare to).

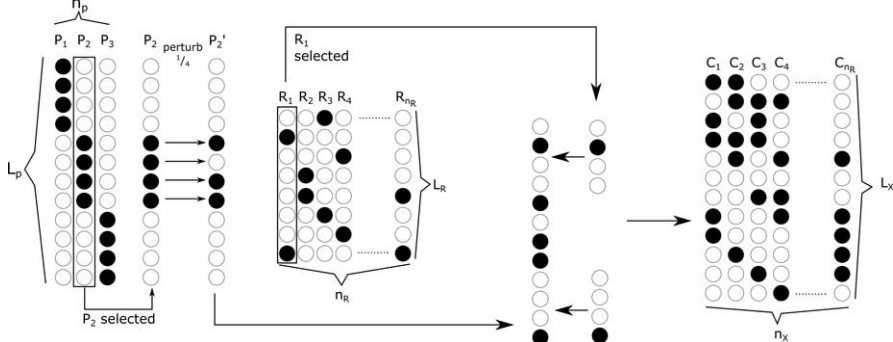

Figure 2: Schematic for building a random code with known properties. Black circles represent ones, white circles represent zeros. Class prototypes are made ($P_1$, $P_2$, and $P_3$) with length $L_P$; the number of prototypes are $n_p$, which is three in this example, their weight is four, with a sparseness number, $S_p$ of $\frac{1}{3}$. Random vectors, $R_x$, are made, as shown, these have length $L_R$ and there are $n_R$ of them; they have weight, $w_R$ of two and sparseness number, $S_R$, of $\frac{1}{4}$. To assemble a new codeword, a prototype is chosen, this example, $P_2$, and 'perturbation' errors are applied, in this example, the perturbation rate is $\frac{1}{4}$, so a single one is turned to a zero in the modified prototype ($P_2^{'}$). A random vector, in this example $R_1$, is then generated, split into blocks and added to the parts of the prototype ($P_1$) that were zero (the random vectors cannot overwrite a random decay in $P$). The process is repeated to create an input 'code' with $n_x$ codewords of length $L_x$, where $n_x = n_R$ and $L_x = L_P$. If the decay values is sufficiently low, members of each class, as they are based on the same prototype are more similar to each other than codewords in other classes.

## 3 EXPERIMENTS

First, we should make that point that finding a single interpretable local code, such as those shown in figure 1, refutes the idea that neural networks do not have interpretable or locally encoded units. The number of local codes is not large, usually between 0-10% of the hidden layer, but this is not insignificant. Furthermore, the number of local codes is tuned by the size the hidden layer, with a peak in the number of local codes seen at $n_{HLN}$=1000 for the standard data set, and a peak in the percentage of HLNs which are local codes seen at $n_{HLN} = 500$ (data not shown): dashed and dot-dashed grey lines are drawn at these points in all relevant figures.

### 3.1 CHANGING THE STRUCTURE OF THE INPUT DATA

Figure 3(left) shows the standard data (black-dashed line) with varying sizes of input data codes ($n_x$ is the number of training examples). More local codes emerge with fewer examples per class, perhaps suggesting that the fewer examples there are, the more efficient it is to learn a short-cut for that class (and the way we set the code up with a prototype block means that there is a short-cut for the neural network to learn). With $n_x = 1000$ the peak is shifted to the left, possibly due to the existence of more different types of solution (see section 3.3.1 for discussion). The effect of sparseness is given in figure 3(right). There are many more local codes when the sparseness is very low, which suggests that the bigger a proportion of the vector is the prototype block, the more drive there is to learn local codes. However, this process is not linear, with the $S_R$ of $\frac{1}{9}$ given very many more codes. Interestingly, the range of Hamming distances between any two members of the same class does not overlap at all with the range of the Hamming distances possible between any two members of any two different classes, whereas, if $S_R$ of $\frac{2}{9}$ or $\frac{3}{9}$ these two sets overlap.

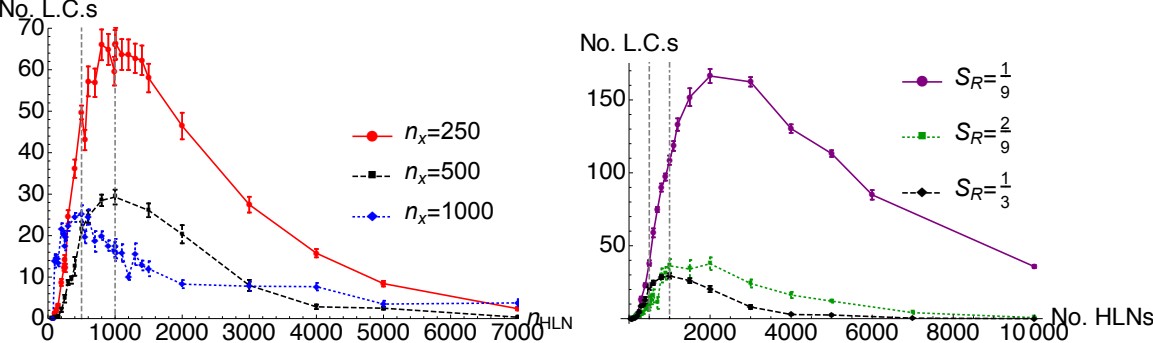

Figure 3: Input data that is few in number and sparse exhibits more local codes. Left: the number of local codes (L.C.s) against the number of HLNs, $n_{HLN}$ for different numbers of training examples ($n_x$). Right: The effect of changing the sparseness of the random blocks of the vector, $S_R$. As the prototype vector in all these cases has a sparseness of $1/10$, the weight of the random $w_R$ and prototype, $w_P$, parts of the codeword and the codewords are 500bits long, note that purple: $S_x = 0.2$, $w_R = 50$, $w_P = 50$; green: $S_x = 0.3$, $w_R = 100$, $w_P = 50$; black dashed: $S_x = 0.4$, $w_R = 150$, $w_P = 50$ Gray dashed and dot-dashed lines are drawn at $n_{HLN} = 500$ and 1000 respectfully.

## 3.2 Changing the Network architecture

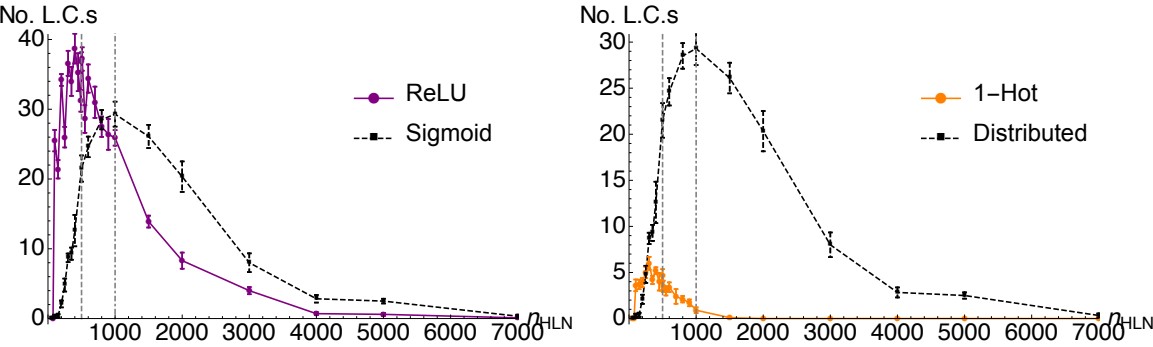

Figure 4: Network architecture parameters can drastically effect the emergence of local codes. Left: Switching from sigmoidal HLNs with a sigmoidal activation function to a rectified linear units (ReLU) neurons; Right: Switching from a distributed to a 1-hot output encoding. Note that the black dashed data is the same as in figure 3. Switching to ReLU gives slightly more LCs, likely due to the fact that ReLUs train quicker and all experiments were stopped after 45,000 epochs. Using a 1-hot output code drastically reduces the need for local codes to emerge in the hidden layer.

Figure 4 shows the effect of changing the HLN activation function and the effect of an 1-hot output encoding. ReLU neurons generally produce a few more LCs, and have a peak shifted to a lower $n_{HLN}$. As all the neural networks were trained for a specified number of steps, this finding may be due to ReLU neurons training quicker (more LCs emerge with longer training, which could be due to LCs being a more efficient solution or more likely due to more proto-selective codes passing the threshold for inclusion).

We chose to use distributed output codes to prevent the possibility that the 1-hot output encoding (which are local codes) would artificially induce local codes in the hidden layer of the network. As shown in figure 4(right), the effect was the opposite with very few local codes emerging if there were a local codes at the output. This suggests that local codes may offer some advantage in a system with distributed inputs and outputs, thus the NN finds them when they are not provided, but that

only a small number is needed, so few are added if they are provided. This suggests that emergent local codes are highly unlikely to be found in the penultimate layer of deep networks.

### 3.3 THE EFFECT OF NOISE

#### 3.3.1 PERTURBATION OF THE PROTOTYPE CODE BLOCK REDUCES LOCAL CODE EMERGENCE

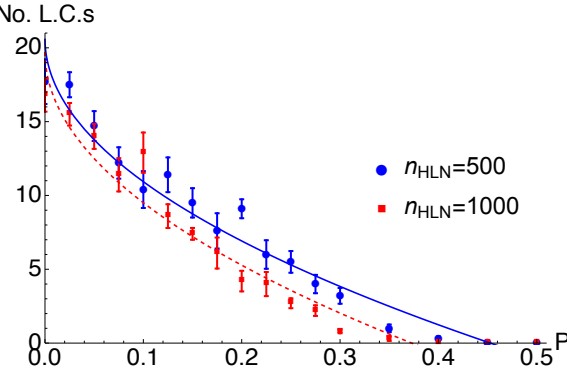

Figure 5: Perturbing the prototype part of the code word decreases the drive to learn local codes. Perturbation, $P$, is measured as the number of bits in prototype block code that are randomly flipped. Data is shown for 500 and 1000 HLNs. No L.C.s emerge when the weight of the prototype block code is twice that of the random blocks.

The random part of the vector already adds some noise on top of the invariant features shared by a class (which is the prototype block), here we demonstrate that the existence of local codes is predicated on there being shared invariant features within a class. Figure 3.3.1 plots the decrease in the number of local codes against the perturbation rate, $P$, where $P$ is defined as the number of bits randomly flipped to zero over the length of the prototype block for the two networks marked with vertical dashed lines in the previous figures. The curves are not well fit by an exponential, the fits shown here are: (HLN=500) 20.62 - 30.65 $\sqrt{P}$, with $R^2 = 0.984$; (HLN=1000) $19.72 - 32.29\sqrt{x}$ with $R^2 = 0.977$, and these data were fit up to $P$=0.4 where the curve crosses zero. Interestingly, the point at which these curves cross the abscissa is the point when the weight of the prototype block is still twice that of the random blocks, i.e. if the random blocks are 33.3% ones, when the prototype block is ≈60% ones, there is no drive to learn any local codes.

This experiment shows that local codes can only emerge (at least in our data here) if there is some factor in common between the categories. This suggests that local codes are unlikely to be found at the very bottom of deep neural networks, where the data has fewer invariant features. However, as neural networks work by re-encoding information in terms of found features, there are class-shared invariants at the higher levels. Furthermore, this does raise the interesting question of whether more local codes will be found in categories that are similar (i.e. a network learning the difference between dog and cat pictures) than in categories that are more random (such as two categories made of random mixes of dog and cat photos), and thus, can the existence of emergent local codes be a measure of true categorical similarity? Or conversely, would a neural network trained on such random data find some invariant features nonetheless? As more LCs were seen in smaller datasets, and irrelevant invariant features are more likely in smaller categories, this suggests that this could be correct.

### 3.4 INCREASING DROPOUT INCREASES THE NUMBER OF LOCAL CODES.

Dropout (see Srivastava et al. (2014)) is a common training technique where a percentage of a layer's neurons are 'dropped out' of the network (their connections are set to zero) during training to prevent over-fitting, however, dropout can also be viewed as a type of training noise. Dropout values of 0%, 20%, 50%, 70% and 90% were applied to the training of the standard network, and the networks were trained repeatedly (220, 310, 260, 250 and 250 times respectively). Results are shown in figure 6 and table 1. There are always solutions with close to zero local codes, but the

Table 1: Various quantities associated with the distribution of local codes (LCs) in with dropout (given as a percentage) applied during training

| No. of LCs | 0% | 20 % | 50% | 70% | 90% |
|---|---|---|---|---|---|
| Minimum | 8 | 4 | 7 | 15 | 0 |
| Maximum | 34 | 29 | 41 | 57 | 125 |
| Mean | 18.44 | 16.13 | 20.65 | 34.54 | 73.80 |
| Standard deviation | 4.55 | 4.19 | 4.96 | 8.05 | 24.53 |

expected (mean) and maximum number roughly increases with an increasing percentage of dropped out neurons. Generally, dropout percentages in the range of 20-50% are used in training, and these probability distribution functions (PDFs), like 0% peak, are also joint peaks, with the 20% data having more of the lower solution and the 50% having more of the higher one. Droping out more than 50% of the network is not generally used as it slows down training. However, with these higher values, the range of solutions is much higher (as evidence by a higher variance and range of the number of local codes), which is expected as dropout forces the network to adopt a range of solution sub-networks, the increase in local codes suggest that localised encoding offers some protection against noise. At first glance, this might seem unlikely, as distributed patterns are claimed to be more resilient against failure. However, say we had a 20% dropout rate, a fully distributed encoding, would be affected by dropout 100% of the time, losing 20% of its information, whereas a localised encoding would be unaffected 80% of the time (although 20% of the time it would lose all data), and further resilience can be provided if duplicate local codes were used for the same class. Note that, as only 10 classes were used, the large number of local codes, especially for the high dropout values, suggests there are multiple LCs for each category. These results suggest that, for a noisy network, solutions involving some duplicate localised codes are useful methods for dealing with uncertainty.

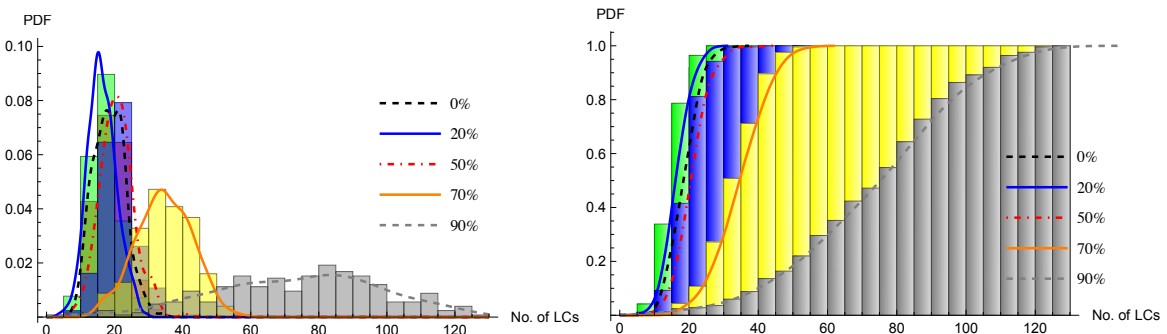

Figure 6: Increasing dropout increases the number of local codes. Probability density function (PDF), left, and cumulative probability density function (CDF), right, of the number of local codes that emerge in repeated neural networks trained with dropout, percentage of the hidden layer neurons dropped out given in the legend. As the dropout percentage increases, generally, the mean and range of local codes found increases, suggesting that localized encoding by the network offers some advantage against noise.

## 4 CONCLUSIONS AND FUTURE WORK

We have demonstrated that interpretable localised encodings of the presence/absence of some categories can emerge from the hidden layer of a feed-forward neural network. As the number of local codes follows a well-defined pattern with the size of the hidden layer, and it is affected by modifications of the input and output data, it suggests that the number of local codes is related to the computing capacity of the neural network and the difficulty of the problem presented to it, suggesting that the local codes offer some modification to the computing power of the neural network.

Furthermore, as the hidden layer size increases, there is so much extra capacity that local codes are not needed. Our results suggest that local codes require more effort to train, but offer more efficient use of the available capacity.

As the number of local codes shared invariants within a categories, it does imply that the local codes have some function associated with recognising these invariants. As the average number and range of LCs generally increases with dropout, and the LCs are repressed by a fully locally encoded output layer, it suggests that some local codes are good to have, and that number increases with noisy networks. The fact that the dropout data seems to contain multiple overlapping peaks, and, in our tests, peak numbers of LCs are seen at 500 and 1000 (and 2000 for the $S_R = \frac{1}{9}$ data) HLNs implies that there are more than one qualitative approaches for the neural network to solve the problem, and tuning the problem and neural network parameters nudges the solution to different distributions of local codes.

Do these simple networks tell us anything about deep neural networks? The data presented here was designed to have invariant feature 'short-cuts' that the neural network could make use of in classifying input data into classes and the argument could well be made that the data passed between layers of a deep neural network is not of the same quality. Whilst an obvious next experiment for us is to investigate the qualities of the data passed within a neural network, preliminary feed-forward neural network training on standard simple neural network data (such as the Iris dataset from Fisher (1936)) results also show the emergence of local codes when there is a 'short-cut' in the data (publication in preparation). The observations that local codes are seen under dropout, with distributed input and output codes, when there are invariant features and local codes are inhibited with 1-hot output encodings, suggests that local codes might be found in a the middle and higher layers of a deep network, and not the penultimate layer where the 1-hot output could inhibit them or the early layers where invariant features common to a class have not yet been identified. Another interesting question is whether the local codes might have a diagnostic use, for example, is it the case that they increased in networks that generalise or are they, perhaps, an indicator of over-training? Answering this would also highlight when and where we should expect invariants in the data, as learning an invariant feature, such as, 'presence of eyes implies presence of face', could help with generalisation and classification, however learning an irrelevant invariant feature, such as, presence of 'blue sky implies tanks' would not. Discriminating a blue-sky selective neuron from a tank-selective neuron in such a case would require careful thought about what we should consider reasonable data to test for selectivity.

## ACKNOWLEDGMENTS

An. Author would like acknowledge funding from Leverhulme on grant no.

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
