# OpenReview forum: "When and where do feed-forward neural networks learn localist representations?"
_ICLR.cc/2018/Conference — Reject_

### Official Review · AnonReviewer2 · 2017-11-22
**Hints of something interesting, but a bit over-simplistic and sloppy.**

**Rating:** 3
**Confidence:** 3

**Review:**

The authors ask when the hidden layer units of a multi-layer feed-forward neural network will display selectivity to object categories. They train 3-layer ANNs to categorize binary patterns, and find that typically at least some of the hidden layer units are category selective. The number of category selective ("localist") units varies depending on the size of the hidden layer, the structure of the outputs the network is trained to return (i.e., one-hot vs distributed), the neurons' activation functions, and the level of dropout-induced noise in the training procedure.

Overall, I find the work to hint at an interesting phenomenon. However, the paper as presented uses an overly-simplistic task for the ANNs, and the work is sloppily presented. These factors detract from my enthusiasm. My specific criticisms are as follows:

1) The binary pattern classification seems overly simplistic a task for this study. If you want to compare to the medial temporal lobe's Jennifer Aniston cells (i.e., the Quiroga result), then an object recognition task seems much more meaningful, as does a deeper network structure. Likewise, to inform the representations we see in deep object recognition networks, it is better to just study those networks, instead of simple shallow binary classification networks. Or, at least show that the findings apply to those richer settings, where the networks do "real" tasks.

2) The paper is somewhat sloppy, and could use a thorough proofreading. For example, what are "figures 3, ?? and 6"? And which is Figure 3.3.1?

3) What formula is used to quantify the selectivity? And do the results depend on the cut-off used to label units as "selective" or not (i.e., using a higher or lower cutoff than 0.05)? Given that the 0.05 number is somewhat arbitrary, this seems worth checking.

4) I don't think that very many people would argue that the presence of distributed representations strictly excludes the possibility of some of the units having some category selectivity. Consequently, I find the abstract and introduction to be a bit off-putting, coming off almost as a rant against PDP. This is a minor stylistic thing, but I'd encourage the authors to tone it down a bit.

5) The finding that more of the selective units arise in the hidden layer in the presence of higher levels of noise is interesting, and the authors provide some nice intuition for this phenomenon (i.e., getting redundant local representations makes the system robust to the dropout). This seems interesting in light of the Quiroga findings of Jennifer Aniston cells: the fact that the (small number of) units they happened to record from showed such selectivity suggests that many neurons in the brain would have this selectivity, so there must be a large number of category selective units. Does that finding, coupled with the result from Fig. 6, imply that those "grandmother cell" observations might reflect an adaptation to increase robustness to noise?

---

> ### Public Comment · (anonymous) · 2017-12-21
> **Simple Constructionist Science, not overly simplistic.**
>
> Thank you for your review and your comments that the work is interesting.
>
> 1. The reviewer stated: ‘The binary pattern classification seems overly simplistic a task for this study. If you want to compare to the medial temporal lobe's Jennifer Aniston cells, then an object recognition task seems much more meaningful, as does a deeper network structure. Likewise, to inform the representations we see in deep object recognition networks, it is better to just study those networks, instead of simple shallow binary classification networks. Or, at least show that the findings apply to those richer settings, where the networks do "real" tasks.’
>
> This is valid, and we are obviously also doing these analyses on deeper networks performing image classification tasks. However, there is value in doing these sorts of classification tasks.
>
> Firstly, if we find localist coding in a deeper and more complex network, how can we possibly know why those codes have appeared if we have not already analysed a much simpler version? This was the motivation behind testing the effects of drop-out and invariance and dataset size independently, so we could use these results to inform our understanding of the deep NN results.
>
> Secondly, our data-set classification task while simple, is not overly simplistic. Using precisely constructed input data with completely known qualities allows us to tease out interactions that would be harder to do. E.g., a criticism that is often applied to the grandmother cell theory and these results is that the patient who is tested is only shown a few hundred items, the number of items in the world that they could recognise is obviously much larger, so even if a cell is found that responds to only one item in the set (i.e. Jennifer Anniston) the argument can always be made that the cell might have responded to an item not shown (like Jeff Goldblum). NNs don’t get tired, so you could show them many different items, but you still cannot show them every item in the world. However, our data sets are designed so that there is a complete (and finite) set of items (i.e. every possible vector of the correct length and patten rules), so we can show the networks every possible item, and thus, find out if the neurons are truly selective.
>
> Thirdly, we designed the input data so that there were invariant parts of the codewords in each category, so there was a short-cut for the NN to learn. This paper has shown how the relative size and structure of these invarients affects the likeliness of NN learning a local code, we can now investigate the structure of the invarients in the representation at the lower levels of the hippocampus to see if there is invarience in representations of category members at that level.
>
> 2. (proof-reading): We will fix all proof-reading issues and typos in the revised version.
>
> 3. (selectivity): The formula for the selectivity is is simply the difference between the highest value activation of one set and the lowest of the other, and the equation has been added to the paper. It was there in words, but perhaps not made clear enough. The reviewer writes: ‘And do the results depend on the cut-off used to label units as "selective" or not (i.e., using a higher or lower cutoff than 0.05)?’ No. I think the best way of thinking about this is the use the word ‘selective’ as a qualitative measure of whether a neuron is selective (to a category) or not, and use ‘selectivity’ for the quantity by which it is selective by, which is the difference in activation between A and not-A groups (An illustration of this was added to figure 1). To answer the question, we found that once the NN had learnt the solve the problem, there were selective neurons, further the training to reduce the error (and move the outputs values closer to 1 and 0) merely increased the selectivity and not the number of neurons. In terms of a direct comparison to single cell recording studies, it isn’t certain which level of selectivity is observable in experiments. The 0.05 cut-off amounts to 5%, which we feel is enough above zero that it is measurably above experimental error. As we could train for longer to increase the selectivity, any cut-off is somewhat arbitrary. However, the number of selective neurons did not change much after the start of training, so the results reported here are valid.
>
> 4. (almost a rant): We shall rewrite the abstract and intro for tone. It wasn’t meant to be almost a rant against PDP, perhaps a little too much excitement slipped into the writing.
>
> 5. (do the noise findings imply that grandmother cells observations might reflect an adaptation to increase the robustness to noise?) Yes! Or rather, we think so. Not being able to do very-long-term evolution experiments with human beings we cannot know for sure. But our approach of looking for general rules of how information is structured in noisy environments does suggest that local codes might be adaptive against excessive noise.

---

### Official Review · AnonReviewer3 · 2017-11-27
**The paper describes a method for determining to what degree individual neurons in a hidden layer of an MLP encode a localist code, which is studied for different input representations.**

**Rating:** 3
**Confidence:** 5

**Review:**

Quality and Clarity
The neural networks and neural codes are studied  in a concise way, most of the paper is clear. The section on data design, p3, could use some additional clarification wrt to how the data input is encoded (right now, it is hard to understand exactly what happens).

Originality
I am not aware of other studies on this topic, the proposed approach seems original.

Significance
The biggest problem I have is with the significance: I don't see at all how finding somewhat localized responses in the hidden layer of an MLP with just one hidden layer has any bearing on deeper networks structured as CNNs: compared to MLPs, neurons in CNNs have much smaller receptive fields, and are known to be sensitive to selective and distinct  features.

Overall the results seem rather trivial without greater implications for modern deep neural networks: ie, of course adding dropout improves the degree of localist coding (sec 3.4). Similarly, for a larger network, you will find fewer localist codes (though this is hard to judge, as an exact definition of selectivity is missing).

Minor issues: the "selectivity" p3 is not properly defined.  On p3, a figure is undefined.
Typo: p2: "could as be".
Many of the references are ugly : p3,  "in kerasChollet (2015)", this needs fixing.

---

> ### Public Comment · (anonymous) · 2017-12-21
> **Basic NN science is significant if we are to understand both brains and deep-NNs**
>
> Thank you for you review, and your comments that our work is original and important.
>
> Regarding your comments on significance, I think perhaps that we have failed to communicate the purpose of our research. Although we are interested in extending this work (in the future) to modern deep neural networks, that is not intended scope for this paper. We want to understand the results of single-cell recording studies in the hippocampus which found possible selective codes. As such, we are trying to elucidate the basic constraints on when such codes appear, and we are doing this by investigating when such codes appear in very simple neural networks with inputs and outputs of known (and easily modifiable) structure, in hope that we can provide hypotheses for when the brain might learn selective codes. Using a modern deep-NN with convolutions and huge number of layers is inappropriate for this task as the higher degree of complexity would obscure the basic constraints on the information representation.
>
> Furthermore, although modern deep neural networks are wonderful (and we are also investigating them), there is a lot of work that needs to be done on the underlying science of why neural networks work the way they do, which is significant for the field, as, although many engineering solutions are found by creatively playing around, there is room for finding engineering solutions by applying the results of basic investigative science (such as is presented basic science we are doing here).
>
> Minor issues and clarity: We fixed the stylistic and clarity concerns in the rewrite.

---

### Official Review · AnonReviewer1 · 2017-11-28
**Interesting idea, intriguing findings, many open questions**

**Rating:** 5
**Confidence:** 4

**Review:**

This paper studies the development of localist representations in the hidden layers
of feed-forward neural networks.

The idea is interesting and the findings are intriguing.  Local codes
increase understandability and could be important for better
understanding natural neural networks. Understanding how local codes
form and the factors that increase their likelihood is critically
important.  This is a good start in that direction, but still leaves
open many questions.  The issues raised in the Conclusions section are
also very interesting -- do the local codes increase with networks
that generalize better, or with overtrained networks?

A  weakness in this paper (admitted by the authors in the
Conclusions section) is the dependence of the results on the form of input
representation.  If we consider the Jennifer Aniston cells, they do
not receive as input as well separated inputs as modeled in this
paper.  In fact the input representation used in this study is already
a fairly localist representation as each 1 unit is fairly selectively
on for its own class and mostly off for the other classes.  It will be
very interesting to see the results of hidden layers in deep networks
operating on natural images.

Please give your equation for selectivity.  On Page 2 it is stated "We
use the word ‘selectivity’ as a quantitative measure of the difference
between activations for the two categories, A and not-A, where A is
the class a neuron is selective for (and not-A being all other
classes)."  However you state that neurons were counted as a local
code if the selectivity was above .05.  A difference between
activations for the two categories of .05 does not seem very
selective, so I'm thinking you used something other than the
mathematical difference.

What is the selectivity of units in the input codewords?  With no
perturbation, and S_x=.2, w_R=50, w_P=50, the units in the prototype
blocks have a high selectivity responding with 1 for all patterns in
their class and with 0 for 8/9 of the patterns in the other classes.
Could this explain the much higher selectivity for this case in the
hidden units?  I would like to see the selectivity of the input units
for each of the plots/curves.  This would be especially interesting
for Figure 5.

It is stated that LCs emerge with longer training and that ReLU
neurons may produce more LCs because they train quicker and all
experiments were stopped at 45,000 epochs.  Why not investigate this
by changing learning rates for one of ReLu or sigmoidal units to more
closely match their training speed?  It would be interesting to see if
the difference is simply due to learning rate, or something deeper
about the activation functions.

You found that very few local codes in the HLNs were found when a
1-hot ouput encoding was used and suggest that this means that
emergent local codes are highly unlikely to be found in the
penultimate layer of deep networks.  If your inputs are a local code
(e.g. for low w_R), you found local codes above the layer of local
codes but in this result not below the layer of local codes which
might also imply (as you say in the Conclusions) that more local
coding neurons may be found in the higher layers (though not the
penultimate one as you argue).  Could you analyze how the selectivity
of a hidden layer changes as a function of the selectivity in the
lower and higher layers?



Minor Note -- The Neural Network Design section looks like it still
has draft notes in it.

---

> ### Public Comment · (anonymous) · 2017-12-21
> **Useful experimental ideas, further explanation of selective and selectivity**
>
> Thank you very much for your detailed and helpful review, it is helpful and enthusiastic reviews like this which are the reason why peer-review is helpful for science. I’m going to go through the questions you raised point by point.
>
> ‘A weakness in this paper (admitted by the authors in the Conclusions section) is the dependence of the results on the form of input representation’ ...’It will be very interesting to see the results of hidden layers in deep networks operating on natural images.’
>
> In honesty, we agree that ‘ It will be very interesting to see the results of hidden layers in deep networks operating on natural images’. I doubt it will surprise you to know that we are also working on finding localist codes in deep neural networks. We have also started to investigate the effects of input representation (and depending on time, I might add in some of these results in the revised paper). However, I think it is worth finding the situations where local codes emerge so we can map when and where they do emerge, and thus, we can start to predict in which sorts of systems we should find them.
>
> Please give your equation for selectivity. On Page 2 it is stated "We use the word ‘selectivity’ as a quantitative measure of the difference between activations for the two categories, A and not-A, where A is the class a neuron is selective for (and not-A being all other classes)." However you state that neurons were counted as a local code if the selectivity was above .05. A difference between activations for the two categories of .05 does not seem very selective, so I'm thinking you used something other than the mathematical difference.
>
> Sorry for not putting in the equation for selectivity. I had written it in in words, but I guess it was not obvious that it was the whole equation. We use the word ‘selective’ as a qualitative measure of whether a neuron is selective (to a category) or not, and use ‘selectivity’ for the quantity by which it is selective by, which is the difference in activation between the disjoint sets of A and not-A, the mathematical difference, as you inferred. However, it is not the size of the difference (the selectivity) that is important. There is no way (that I can think of, anyway) to directly relate these selectivity values to outputs from (living) neurons, so the exact numerical value is largely irrelevant (and can be increased by further training, see my response to reviewer 2). A better metric is whether the neuron is selective or not. For example, we had 500 codewords separated into 10 classes. So if the activations were distributed randomly, the chances of all the members of one class ending up higher (or equivalently, lower) than the other classes is (50 choose 50) / (500 choose 50) which is tiny (4.32*10^-71). So if all the members of one class have activations which are disjoint from the set of all activations, then the neuron is selective. If the gap between the set of all A and the set of all not-A is a measure of how long the NN has been trained, then the numerical value of the selecitvity is less important than simply finding a selective unit.
>
> To answer the question, we found that once the NN had learnt the solve the problem, there were selective neurons, further the training to reduce the error (and move the outputs values closer to 1 and 0) merely increased the selectivity and not the number of selective neurons. In terms of a direct comparison to single cell recording studies, it is not certain which level of selectivity is observable in experiments. The 0.05 cut-off amounts to 5%, and we felt was reasonably enough above zero that it could count as measurable above experimental error. However, as I said, we could train for longer to increase the selectivity, and thus, any cut-off is somewhat arbitrary. However, the number of selective neurons did not change much after a short amount of training, so the results reported here (i.e. the number of selective units) are valid. I have added further explanation of the measure of selectivity to the paper and figure 1, and added discussion of finding selective units.
>
>
>
> It is stated that LCs emerge with longer training...’ ‘Why not investigate this by changing learning rates for one of ReLu or sigmoidal units to more closely match their training speed? It would be interesting to see if the difference is simply due to learning rate, or something deeper about the activation functions.
>
> This is an interesting idea. We are going to look at stopping both the ReLu and sigmoidal neuron cases at the exact same loss and then comparing. But we could investigate the learning rate as well.
>
> I’ve posted this response up now, in case you would like further discussion. And I will try and run all the suggested experiments by the deadline, and I shall post the results up here as I get them.

---

> ### Public Comment · (anonymous) · 2017-12-21
> **Input codes are not as selective as one might think**
>
> You stated:
> 'What is the selectivity of units in the input codewords? With no perturbation, and S_x=.2, w_R=50, w_P=50, the units in the prototype blocks have a high selectivity responding with 1 for all patterns in their class and with 0 for 8/9 of the patterns in the other classes. Could this explain the much higher selectivity for this case in the hidden units? I would like to see the selectivity of the input units for each of the plots/curves. This would be especially interesting for Figure 5.'
>
> We have already started to investigate how changing the input patterns affects the number of local codes seen in the NN. It is an interesting idea to change the selectivity, by, say, using the numerical input of 0.3 to stand for 0 and 0.7 to stand for 1, I could then look and see if, after a required amount of training, if the size of the selectivity is smaller. If I have time to run this, I will add it to the revised paper.
>
> However, the input codes do not have very many selective units. For each of the 10 classes, we have a prototype part of the code which is 50 bits long, leaving 450 bits which are randomly assigned to be 1 with a probability of 1/3. So, for a unit to be selective in the unperturbed case, 450 input codes would have to all have a zero for that neuron, which is the chance of a zero coming up 450 times, and the chance of a single zero is 2/3. The overall chance of a unit being selective, then, is very small. To check, and because at first glance I thought that number of selective input units would vary for the data in figure 5, and this might explain some of the trend shown, I wrote some code to count the number of selective units. However, in my test case, I got 0 units (for the data with a sparseness of 0.4). As the reviewer specifically mentions the data with S_x of 0.2, I tested that as well, and also got zero selective codes.

---

### Public Comment · (anonymous) · 2018-01-05
**List of changed made as the paper was revised**

What was done in the rewrite of the paper:

1. Corrected typo’s, references, etc
2. Rewrote introduction for clarity and tone, and out-lined the motivation more clearly (as I think we didn’t explain it well enough and one of the reviewers misunderstood our motivation)
3. Added in the formula for selectivity (as suggested by the reviewers), and an argument for why the presence of a selective neuron is more important than the amount it is selectivie by (as the equation wasn’t originally included to avoid giving the misleading impression that the quantitative selectivity was more important than the emergence of selective codes).
4. Added in the measure of selectivity in fig 1, to make the formula more clear
5. Added in the explanation for why we used such a simple task at the end of the intro, as this point was questioned by a reviewer.
6. Stated that local codes were highly unlikely to be present in the input code (as this question was asked by a reviewer)

Most the changes are in in the rewrite of the introduction to increase clarity, the results and their write-up are unchanged from the first version.

---

### Decision · Program_Chairs · 2018-01-29
**ICLR 2018 Conference Acceptance Decision**

**Decision:**

Reject

**Comment:**

This work looks at what factors can lead to the emergence of selectivity (to certain categories) in units of a neural network. While this is an intriguing area to explore, this work uses settings that are quite toy-ish, making it a very hard to see how the observations could generalize to more realistic architectures or tasks.